# Characterization of the Caudal Ventral Tubercle in the Sixth Cervical Vertebra in Modern *Equus ferus caballus*

**DOI:** 10.3390/ani13142384

**Published:** 2023-07-22

**Authors:** Sharon May-Davis, Diane Dzingle, Elle Saber, Pamela Blades Eckelbarger

**Affiliations:** 1Canine and Equine Research Group, University of New England, Armidale, NSW 2351, Australia; 2Equus Soma—Equine Osteology and Anatomy Learning Center, Aiken, SC 29805, USA; dldzingle@gmail.com (D.D.); info@equus-soma.com (P.B.E.); 3Biological Data Science Institute, Australian National University, Canberra, ACT 2601, Australia; elle.saber@anu.edu.au

**Keywords:** congenital malformation, cranial ventral tubercle, equine complex vertebral malformation, *longus colli*, homeotic transformation, hox, ventral process

## Abstract

**Simple Summary:**

Specialized bony structures in the cervical spine are conservative traits in *Equus ferus caballus* with the ventral process of C6 being deemed one of the most significant structures relative to posture and locomotion. However, studies have identified an anomalous variation in the ventral process of C6 where its caudal ventral tubercle (CVT) is either unilaterally or bilaterally absent (aCVT). Consequently, the aim of this study is to describe the morphology and provide a grading system that identifies the extent of the absence in equal quarterly increments. Only osseous specimens with congenital variants of the ventral process were examined, totaling 76 (unilateral *n* = 47; bilateral *n* = 29). Observational grading identified four levels of absence—1/4, 2/4, and 3/4, with 4/4 representing complete absence of the CVT. Fifty-six osseous specimens with aCVTs had a grade 4/4 with 41/56 extending the absent morphology cranial, thus involving the caudal aspect of the cranial ventral tubercle (CrVT). This combined morphology effectively diminished the attachment sites for the cranial, and thoracal portions of the *longus colli* muscle, a primary fixator, flexor, rotator, and stabilizer of the cervical vertebrae. A study comparing radiographic images with gross specimens may provide information to facilitate accurate radiological descriptions of C6 variants.

**Abstract:**

This study examined the anomalous variations of the ventral process of C6 in modern *E. ferus caballus.* The aim was to provide an incremental grading protocol measuring the absence of the caudal ventral tubercle (CVT) in this ventral process. The findings revealed the most prevalent absent CVT (aCVT) was left unilateral (*n* = 35), with bilateral (*n* = 29) and right unilateral (*n* = 12). Grading was determined in equal increments of absence 1/4, 2/4, 3/4, with 4/4 representing a complete aCVT in 56/76, with a significance of *p =* 0.0013. This also applied to bilateral specimens. In those C6 osseous specimens displaying a 4/4 grade aCVT, 41/56 had a partial absence of the caudal aspect of the cranial ventral tubercle (CrVT). Here, grading absent CrVTs (aCrVT) followed similarly to aCVTs, though 4/4 was not observed. The significance between 4/4 grade aCVTs and the presentation of an aCrVT was left *p =* 0.00001 and right *p =* 0.00018. In bilateral specimens, C6 morphologically resembled C5, implying a homeotic transformation that limited the attachment sites for the cranial and thoracal *longus colli* muscle. This potentially diminishes function and caudal cervical stability. Therefore, it is recommended that further studies examine the morphological extent of this equine complex vertebral malformation (ECVM) as well as its interrelationships and genetic code/blueprint.

## 1. Introduction

Segmentation and specific osteological features of the axial skeleton are a conserved trait in most mammals, as demonstrated in the fossil record and extant specimens [1,2,3,4,5,6,7,8,9]. One of the largest examples of complete fossil specimens belongs to the Family Equidae with a 55-million-year history of ancestral equids. Even today, extant equids have similar examples of specific morphological traits that have been evident throughout evolution and especially in the cervical vertebrae, where atypical morphologies were noted [5,6,7].

In modern *Equus ferus caballus*, cervical vertebrae demonstrate typical and atypical morphology, similar to ancestral equids [5,6,7,8,9,10,11]. Typical cervical vertebrae consist of basic structures shared with most other vertebrae and no specialized osteological features, notably C3–C5; while atypical cervical vertebrae (C1, C2, C6, and C7) have specific structural modifications that are not shared with other vertebrae. These specialized vertebrae are often defined by position and function, and according to Arnold (2021), C6 in the caudal cervical module (C5–C7) is biomechanically significant in supporting posture and head/neck function [1,2,7,8,9,10,11]. Located in the cervicothoracic junction (C5–T2), the specialized morphology of C6 in *E. ferus caballus* is characterized by bilateral ventral projections of bony sheets that traverse the entire length of the vertebral body. This structure is referred to by several names in the literature, ventral tubercle, ventral lamina (*lamina ventralis*), and/or ventral transverse process [9,10,11,12,13,14,15,16,17]. For this study, it will be referenced as the ventral process of C6.

In *E. ferus caballus,* these bilateral bony projections in C6 present tube-like morphology along the lateral ventral border in a craniocaudal orientation, separate and distal to the transverse process [7]. From the left lateral view, each ventral process has a CrVT cranial to the transverse process while the CVT is caudal. These two tubercles are separated by a small convexity directly distal to the transverse process. From the ventral view, the CrVT and CVT are separated by a small constriction that is in alignment with the transverse process; the CVT appears wider than the CrVT, while the overall outside width is greater across the CrVTs [7]. Rombach et al. (2014) describe the CrVT in *E. ferus caballus* as the attachment site for the cervical portion of the multi bundled *longus colli* muscle, while the CVT is the insertion point for the thoracal tendon of the *longus colli* muscle that extends caudally to its origin on either T5 or T6 [18]. Additionally, the thoracal tendon itself remains within the muscle belly from its attachment at C6 to T2 and acts as a ventral support for the directional transition of the vertebrae in the cervicothoracic junction [18]. Functionally, the cervical and thoracal portions of the *longus colli* muscle aid in the fixation, stabilization, rotation, and flexion of the cervical vertebrae, while Bainbridge describes it as a site of force redirection during muscle contraction cranial to and/or caudal to C6 [11,18].

Studies examining the CVT of C6 reveal a congenital malformation in modern *E. ferus caballus* and described the aCVT in multiple breeds of horse from different geographical populations [15,16,17,19,20,21,22,23]. This aCVT can have either unilateral or bilateral morphological change; however, some case reports have identified further associative congenital malformations directly caudad to C6. These involve cervical and thoracic vertebrae, sternal ribs (1st and 2nd), muscles (*longus colli* and *scalenus ventralis*), nerves (brachial plexus and phrenic), and the trachea [15,19,20,21,22,23]. Moreover, these anomalous variations of C6 have not been identified in extinct *Equus*, nor sister taxa of *E. ferus caballus* [7]. In Holstein cattle, a similar congenital malformation influenced by a missense mutation has been described [24,25,26]. This syndrome is referred to as complex vertebral malformation (CVM) and includes multiple axial skeletal variants in or near the cervicothoracic junction, including anomalous sternal ribs [24,25]. *Equus ferus caballus* presents similar gross morphologies to that described in Holstein cattle (CVM), with no comparative genetic research [15,16,17,19,20,21,22,23]. Therefore, as both species display similar gross morphologies, the equine equivalent could be described in terms of an equine complex vertebral malformation (ECVM). However, one important aspect of ECVM not reported in CVM is that an aCVT of C6 is evident in *E. ferus caballus* [15,16,17,19,20,21,22,23].

To date, the aCVT of C6 has not been fully examined or described, nor the various morphological presentations reported in *E. ferus caballus*. Therefore, the primary objective of this observational study is to characterize and establish the variations of the aCVT of C6 and provide a grading system relevant to the extent of the absent morphology. Although clinical findings are beyond the scope of this study, the conclusions might facilitate the clinician’s description of the anomalous variations in C6 when radiographed.

## 2. Materials and Methods

### 2.1. Ethical Statement

No horses were euthanized for the purpose of this study and observational research was conducted postmortem from osseous specimens of C6 with owner permission of private collections, educational facilities, and one private research facility.

### 2.2. Terminology

Primary nomenclature was derived from Sisson [8], Getty [9], and previous literature describing the aCVT of C6 [15,16,17,19,20,21,22,23].

### 2.3. Materials

To be eligible for the study, only osseous specimens of C6 from modern breeds of *E. ferus caballus* that had an aCVT with minimal damage and clear structural definition of the remaining ventral process were selected for examination.

Six facilities housed suitable osseous specimens and granted access to their collection/s—three private, one private research facility, one College, and one University (outlined in acknowledgements).

One normal C6 was represented by a 15-year-old male Oldenburg × Thoroughbred from the United States of America (USA) for comparative purposes. Seventy-six C6 osseous specimens with aCVTs in modern *E. ferus caballus* were examined, totaling 152 CVTs. The suitable C6 specimens were sourced from nine countries: Australia (*n* = 44); USA (*n* = 17); Japan (*n* = 4); the Netherlands (*n* = 3); New Zealand (*n* = 3); United Kingdom (*n* = 2); Belgium (*n* = 1); Ireland (*n* = 1); Sweden (*n* = 1). Nine breeds (not inclusive of Crossbreds) were represented by Thoroughbreds (*n* = 40); Warmbloods (*n* = 13); Australian Stock Horses (*n* = 6); Standardbreds (*n* = 4); Appaloosas (*n* = 3); Quarter Horses (*n* = 2); Friesian (*n* = 1); Irish Sport Horse (*n* = 1); Riding Pony (*n* = 1); and Crossbreds (*n* = 5). The age by year ranged from stillborn to 30. The study consisted of 47 males and 29 females. Individual details are reported in Appendix A.

Photographs were acquired using a NEEWER^®^ White Soft Box 32 × 32 inch (81.3 × 81.3 cms) photobooth with a Canon EOS RP camera and 50 mm focal length lens. The camera was fixed to a NEEWER^®^ Mini Travel Tabletop Tripod with a height range of 24 inches (60.9 cms) above normal desk height. Four NEEWER^®^ T120 LED lights set at 5400 Kelvin were utilized for lighting of the specimens. Only ventral and left lateral photographs were acquired for the study.

### 2.4. Methods

From the ventral view, the normal C6 with the caudal aspect of the CVT in alignment with the vertebral body, we standardized the demarcation of the ventral process into CrVT and CVT by identifying the unification of the caudal border of the transverse process with the vertebral body as our point of reference. The width of the CrVT and CVT were determined by the visual examination of the enthesis patterns (Figure 1).

From the ventral view of a normal C6, with the caudal aspect of the CVT in alignment with the caudal aspect of the vertebral body, the ventral process is divided into the CrVT and CVT as per Figure 1. Then, for grading purposes, the two divisions (CrVT and CVT) per ventral process are divided equally into quarterly increments for ease of conceptualization and graded 1/4, 2/4, 3/4, and 4/4 from caudal to cranial (Figure 2).

Each grade indicates the extent of the absence per tubercle. As the grading incrementally increases, so does the relationship to the extent of the absence. For example, when an aCVT displays a 1/4 grade, it indicates one-quarter of the CVT is caudally absent; a 2/4 grade indicates one-half of the caudal CVT is absent, and so on. When an aCVT has a 4/4 grade, then the entire CVT is absent. The assessment process was determined through comparative visual observation as left to right bony landmarks were quite variable among the 76 osseous specimens. When the CVT was completely absent, care was taken to examine a possible caudal absence of the CrVT and the same incremental grading protocols were applied to the CrVT (Figure 3).

Observations of the C6 osseous specimens were conducted by 3/4 authors: Sharon May-Davis, Diane Dzingle, and Pamela Blades Eckelbarger (concurrently 17/76; independently SM-D 59/76).

The aCVT grades from 1/4 to 4/4 can be seen in Figure 4.

### 2.5. Statistical Analysis

Data processing, statistical analysis, and visualizations were completed using R version 4.2.2 (R Core Team 2022) [27], the tidyverse packages [28] and the gtsummary package [29]. A chi-squared test was used to test dependence between variables, or Fisher’s Exact Test where cell counts were small.

These were used to determine the relevance of origin (country/continent), breed (purpose of breed; racing Tb, St, and QH, or riding), age, sex, and associative patterns of tubercle loss between left and right ventral processes.

## 3. Results

Seventy-six osseous specimens (152 CVTs) of C6 had the following absent morphology—bilateral (*n* = 29) and unilateral (*n* = 47) equating to 105/152 aCVTs.

### 3.1. Individual Grades

Individual grades of the aCVTs are reported in Appendix A; the combined summary can be seen below in Table 1.

When an aCVT had a 4/4 grade, the absent morphology could extend along the ventral process cranial to the transverse process and include the caudal aspect of the CrVT. This inclusive morphology of a partially aCrVT was graded in equal increments of 1/4, 2/4 and 3/4, with 4/4 being a completely aCrVT (not reported), as shown in Figure 5.

### 3.2. Statistical Analysis: Absent CVTs by Morphology—Bilateral or Unilateral

Of the 76 osseous specimens specifically selected for aCVTs, a unilateral absence was the most prevalent (*n* = 47; 61.8%), with bilateral (*n* = 29; 38.2%). The left unilateral aCVT was the most prevalent morphological presentation in the sample (*n* = 35; 46.1%). The average age of the horse at the time of death was 11.3 years (median 10; interquartile 6,16).

Tests for independence between the type of anomalous variation and the country/continent of origin, breed (purpose of breed) and gender showed no evidence of an association. Furthermore, there was no statistically significant difference between the mean age groups (Table 2).

The distribution of origin, breed (purpose) and sex are readily viewed in Figure 6.

Of the 76 osseous specimens with aCVTs of C6, 56/76 had a 4/4 grade. This grade presented the highest percentages in both left and right CVTs with a Fisher’s Exact Test noting a *p*-value = 0.0013 between left and right aCVTs (Table 3).

### 3.3. Statistical Analysis: Absent CrVT Patterns of Loss

Absent CrVTs were noted in 41/76 (53.9%) osseous specimens with grades between 1/4–3/4; none had a 4/4 grade. The aCrVT had bilateral and unilateral (left 37/41 and right 17/41) morphology. The presentation of aCrVTs only occurred when the aCVTs graded 4/4 (complete absence of the CVT). The strength of this dependence between the aCVT and aCrVT is demonstrated by highly significant p-values using a Fisher’s Exact Test of Independence (left *p* = 0.00001 and right *p* = 0.00018). There is strong evidence of an association between a 4/4 grade aCVT and aCrVT (Table 4).

### 3.4. Bilateral Presentations

Twenty-nine C6 osseous specimens had a bilateral aCVT 4/4 grade; in 24, the caudal aspect of the CrVT was also absent. Therefore, C6 had a similar shape to C5. In normal C5 morphology, the CrVT combines with the transverse process at its cranial border, forming a continuance between the two structures. This morphological feature can also be seen in the bilaterally aCVT of C6 (Figure 7).

In the 29/76 bilaterally aCVTs, 24/29 had bilateral 4/4 grades (24/76: 32% overall) with 19/29 aCrVTs—unilateral 6/19 and bilateral 13/19. As noted in bilateral aCVTs, bilateral aCrVTs can also have asymmetrical gradings (Figure 8).

## 4. Discussion

This study established a grading system in 76 osseous specimens that evaluated the absent CVT in C6 further to existing literature [9,15,16,17,19,20,21,22,23,30], using a system similar to Haussler et al [31]. It adopted quarterly increments comparative to the degree of bony absence with findings revealing morphological variations that to date have not been reported. Although aCVTs have already been well established via radiographic images [32,33], a study comparing radiographic images with gross specimens might provide information that facilitates accurate radiological descriptions of C6 variants.

Although asymmetric findings were common in this study, morphological asymmetry between left and right expressions during the development of visceral organs in vertebrates is an integral part of the body plan [34,35]. Even so, erroneous deviations from normal morphology can impede evolutionary fitness [36] and according to May-Davis, Hunter, and White [7], anomalous variations of the CVT or CrVT in the ventral process of C6 were not evident in ancestral *Equus*. This concurs with the overall findings in the fossil record where congenital malformations are considered rare [37]. Nonetheless, the fast and conscious shift in the selection of diversified breeds with differing phenotypes, coupled with effective inbreeding programs, has come with deleterious consequences, both genetically and phenotypically [38,39,40,41]. Some of these deleterious mutations are breed and/or color oriented such as hyperkalemic periodic paralysis in Quarter Horses [42] and congenital stationary night blindness in spotted horse breeds [43]. DeRouen et al. (2016) postulated the occurrence of aCVTs in differing breeds and populations suggests a genetic basis [16]. This study concurs by reporting aCVTs from varying breeds in both hemispheres. Additionally, Santinelli et al. (2016) described a propensity towards aCVTs in females [17], while the findings here agree with other studies that neither sex is significant [15,16,19,20,21,22,23]. Several studies associated aCVTs with the likelihood of intravertebral sagittal ratios less than normal (<0.5), and a positive association between perceived cervical pain and/or decreased range of motion (ROM) [16,21]. In contrast, Veraa et al. (2019) found no clinical relevance of aCVTs in Warmblood horses [20].

The functionality of specialized cervical vertebrae and associative musculature is well documented [1,2,18,44,45,46,47,48,49,50] and when not resting, Equidae, as per most mammals, holds the cervical spine as vertically as possible to reduce the distance between the weight of the head and the sustaining cervicothoracic junction [1,2]. This leads to the stereotypical vertical, s-shaped, and self-stabilizing resting posture of the cervical spine [51,52]. As head/neck movements start from this posture, changes in orientation and awareness, such as gazing, are along the sagittal plane. These movements are therefore restricted to the cranial cervical module (occiput to C2) and the caudal cervical module [1,2]. Arnold (2021) states this functional modularity is supported by two prominent bony processes that provide major muscle attachment sites for head and neck motion: the enlarged spinous process on C2, and the ventral process of C6 [1]. These findings support Bainbridge’s (2018) comments that the ventral process of C6 acts as a site of force redirection during muscle contraction cranial to and/or caudal to C6 [11]. Furthermore, one study identified the C5–C6 articulation as a ‘low-motion high-pressure’ joint because the heavy head and neck segment is levered from the caudal cervical spine [50]. Biomechanically, this would assist the highly mobile caudal cervical module and cervicothoracic junction in posture and locomotion combined with the hypaxial actions of the *longus colli* muscle. Hence, any impediment to supportive structures such as a unilateral or bilateral aCVT impacts upon the attachment sites of the thoracal *longus colli* muscle and therefore the structural morphology as noted in the thoracal portions reduced tendon morphology and hypertrophy of the muscle belly when the CVT is absent [23]. This structural deficit quite possibly explains some of the caudal neck pain and limited ROM reported by Beccati et al. (2020) [21] and DeRouen et al. (2016) [15], in contrast to Veraa et al. (2019) [20]. Even so, the high prevalence of caudal cervical osteoarthritis could also be a contributary factor [31].

In previous studies, congenital malformations coinciding with aCVTs of C6 include transposition of the aCVT from C6 onto the ventral surface of C7, anomalous 1st and 2nd ribs, variations in soft tissue attachments and hypertrophy, and neural variations [15,19,22,23,33]—hence, the collective acronym ECVM. As CVM originated from two closely related bulls [25,26], similar concerns apply to the inbreeding programs of *E. ferus caballus* and its association with the increase in deleterious mutations [38,39,40,41]. Even now, certain breeds are being challenged, with several nearing extinction [42,43,53,54]. Furthermore, in those cases where the transposition of aCVTs onto the ventral surface of C7 are reported, this altered morphology is referred to as a “homeotic transformation”. This occurs when either a partial or complete change in the identity of one vertebra transforms into the likeness of a neighboring vertebrae [55]. In the anterior to posterior axis of the developing skeleton, the homeobox family of transcription factors, the Hox genes, in conjunction with a multitude of regulatory factors, are essential for the specification of vertebral identities [56,57,58]. Additionally, experimental studies in mice have shown that targeted disruptions in Hox genes result in homeotic transformations of the cervical vertebrae [59,60]. Of particular interest are the morphological descriptions of these transformations in mice, which are similar to that evidenced here in this study of modern *E. ferus caballus*. Additionally, it was shown in mice that the Hoxa5, Hoxb5, and Hoxb6 mutants as well as the triple Hoxa4/Hoxb4/Hoxd4 mutants were responsible for missing and transposed ventral tubercles [57,61,62,63,64]. It is important to note that when homeotic transformations have been observed in previous studies with bilateral transposition of the aCVTs from C6 onto the ventral surface of C7 [15,19], the mammalian body plan of seven cervical vertebrae is unchanged. However, the size of the aCVT has not been correlated to the frequency, or the size of the transposition onto the ventral surface of C7, nor the variations in musculature. Therefore, further research might reveal similar genetic mutations, as per the targeted murine studies, are occurring in modern *E. ferus caballus*, where the orchestrated disruptions in the normal expression of one or more Hox genes resulted in homeotic transformations.

Finally, limitations to the current study involved locating osseous specimens with aCVTs; hence, a study comparing radiographic images to gross specimens might facilitate further studies in live specimens.

## 5. Conclusions

Seventy-six osseous specimens of C6 with aCVTs were examined and a grade protocol established that revealed the most prevalent aCVT finding was the left unilateral (*n* = 35), then bilateral (*n* = 29) with right unilateral (*n* = 12). The most consistent aCVT grade was 4/4 (56/76) with the absence extending to include the CrVT in 41/76. In bilateral 4/4 graded aCVTs, C6 resembled C5, implying homeotic transformation. Therefore, it is a recommendation of this study to gather further information regarding the anomalous variations associative with the aCVT of C6 (genetic and morphological) and whether lesser or greater grades of aCVTs are conducive of transposition onto the ventral surface of C7.

## Figures and Tables

**Figure 1 animals-13-02384-f001:**
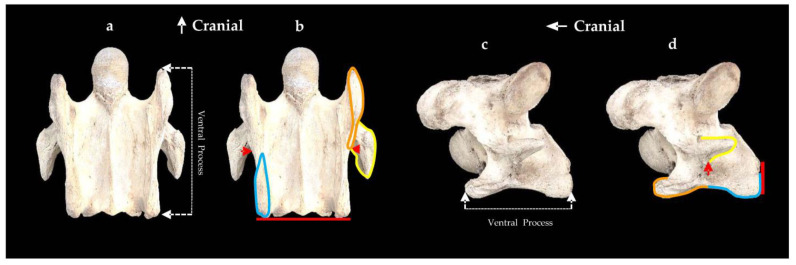
The normal ventral and left lateral view of the sixth cervical vertebra in a 15-year-old Oldenburg × Thoroughbred male. (**a**) Ventral view of the ventral process. (**b**) The red line denotes the level between the caudal ventral tubercles (blue outline) and vertebral body; the red arrows identify the caudal border of the transverse process uniting with the vertebral body; the orange line denotes the cranial ventral tubercle; the yellow line determines the outline of the transverse process. (**c**) Lateral view of the ventral process. (**d**) The red arrow denotes the unification of the caudal aspect of the transverse process with the vertebral body; the caudal ventral tubercle (blue outline) is in alignment with the vertebral body (red line); the transverse process is outlined with a yellow line; the orange line denotes the cranial ventral tubercle.

**Figure 2 animals-13-02384-f002:**
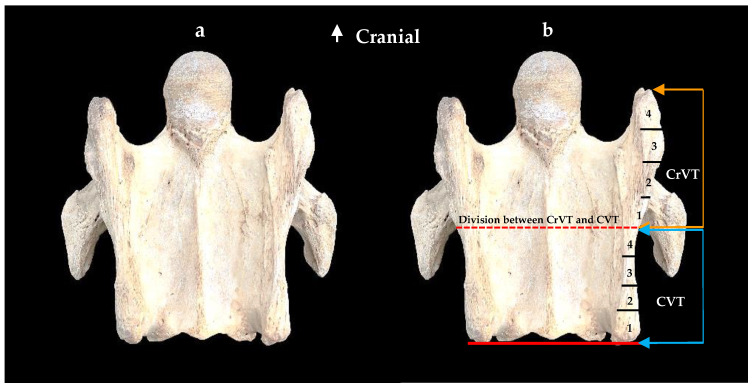
The normal ventral view of the sixth cervical vertebra in a 15-year-old Oldenburg × Thoroughbred male. (**a**) Ventral view. (**b**) Red dotted line indicates the demarcation between the cranial (orange arrows) ventral tubercle and the caudal (blue arrows) ventral tubercle; each tubercle presents four equally incremental graduations from 1–4.

**Figure 3 animals-13-02384-f003:**
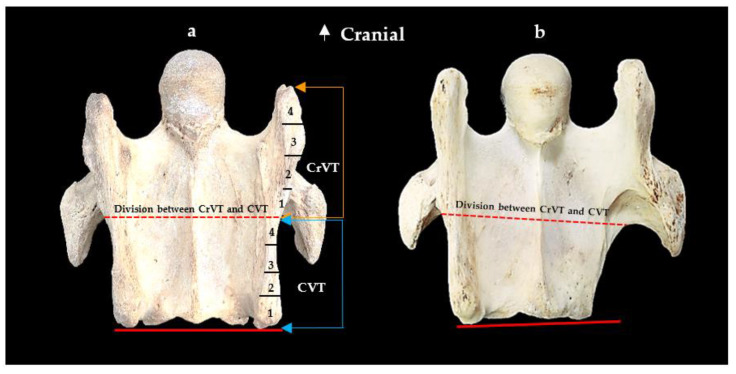
The ventral view of the sixth cervical vertebra. (**a**) A normal sixth cervical vertebra showing the graduations of the caudal (blue arrows) and cranial (orange arrows) ventral tubercles. (**b**) The ventral view of a left absent caudal ventral tubercle with the red dotted line indicating the demarcation between the cranial ventral tubercle and the caudal ventral tubercle; this presentation represents a grade 4/4 absent caudal ventral tubercle and grade 1/4 absent cranial ventral tubercle.

**Figure 4 animals-13-02384-f004:**
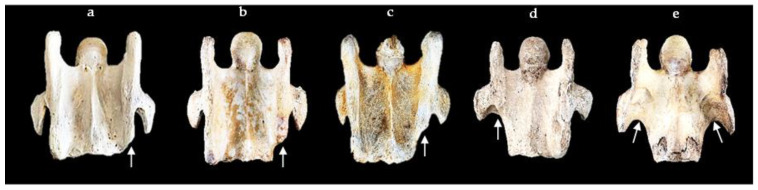
The grading from 1/4–4/4 of unilateral and bilateral absent caudal ventral tubercles (white arrows). (**a**) Tb 29—unilateral left grade 1/4. (**b**) Tb 4—unilateral left grade 2/4. (**c**) Wb 3—unilateral left grade 3/4. (**d**) ASH 3—unilateral right grade 4/4. (**e**) Fr 1—bilateral grade 4/4 left and right.

**Figure 5 animals-13-02384-f005:**
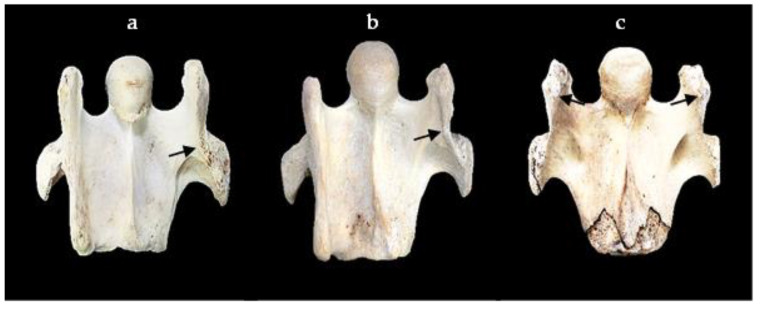
The grading from 1/4–3/4 of unilateral and bilateral absent cranial ventral tubercles (black arrows). Each specimen also had a coinciding 4/4 grade absent caudal ventral tubercle. (**a**) Tb 27—unilateral left grade 1/4. (**b**) ASH 4—unilateral left grade 2/4. (**c**) ASH 2—bilateral grade 3/4.

**Figure 6 animals-13-02384-f006:**
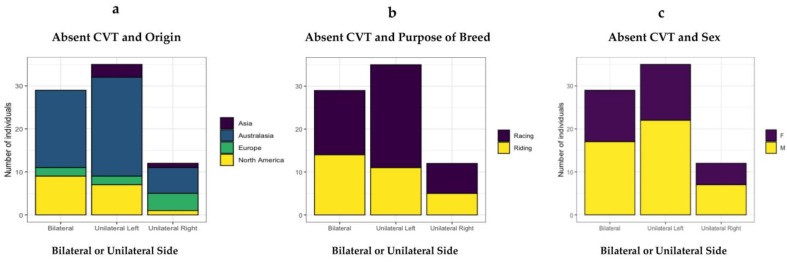
The distribution of absent caudal ventral tubercles in 76 horses. (**a**) Origin by country/continent. (**b**) Purpose of breed—racing or riding. (**c**) Sex—male or female.

**Figure 7 animals-13-02384-f007:**
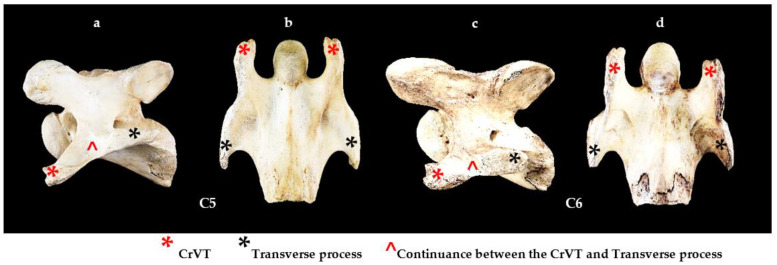
Left lateral and ventral view of a normal C5, and bilaterally absent caudal and cranial ventral tubercles of C6 from Fr 1. (**a**) Left lateral view normal C5. (**b**) Ventral view normal C5. (**c**) Left lateral view bilaterally absent caudal and cranial ventral tubercles C6. (**d**) Ventral view bilaterally absent caudal and cranial ventral tubercles C6. NOTE: In (**c**), the cranial ventral tubercle (red asterisk) fuses with the cranial border of the transverse process (black asterisk); this presentation of C6 is similar to a normal C5 (**a**).

**Figure 8 animals-13-02384-f008:**
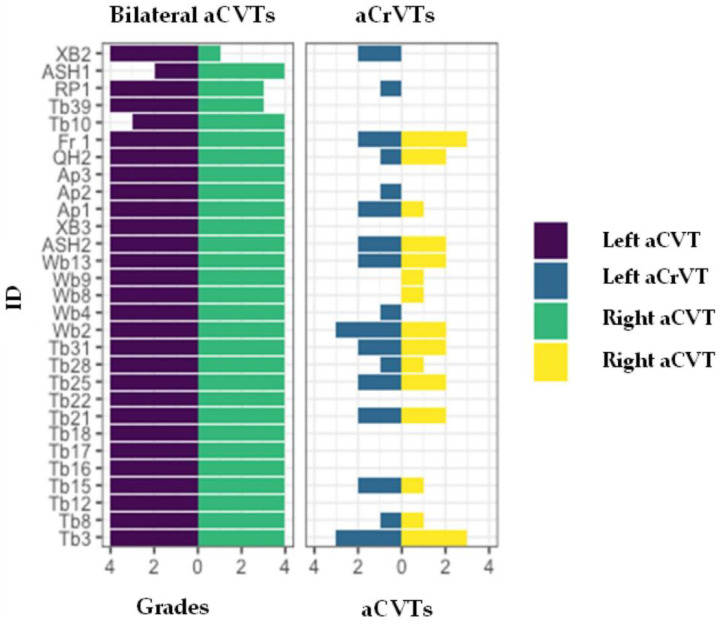
Correlating grades of the C6 osseous specimen between 29 bilaterally absent caudal ventral tubercles and corresponding absent cranial ventral tubercles.

**Table 1 animals-13-02384-t001:** Summarized grading results from 76 osseous specimens of the sixth cervical vertebrae selected for anomalous variations of the ventral process.

Laterality	Grade	aCVT Left	aCVT Right	aCrVT Left	aCrVT Right
Unilateral	1/4	3	1	4	
	2/4	5	4	14	2
	3/4	5	1	2	
	4/4	22	6		
Bilateral	1/4		2	7	6
	2/4	1		8	6
	3/4	1	1	4	3
	4/4	27	26		

Key: a—absent.

**Table 2 animals-13-02384-t002:** Statistical analysis of the 76 sixth cervical vertebrae selected for anomalous variations of the ventral process detailing origin, breed (purpose), average age, and sex.

Characteristic	Bilateral, *n* = 29	Unilateral Left *n* = 35	Unilateral Right *n* = 12	*p*-Value
Origin				0.0787 ^1^
Asia	0 (0%)	3 (8.6%)	1 (8.3%)	
Australasia	18 (62%)	23 (66%)	6 (50%)	
Europe	2 (6.9%)	2 (5.7%)	4 (33%)	
North America	9 (31%)	7 (20%)	1 (8.3%)	
Breed (purpose)				0.3970 ^2^
Racing	15 (52%)	24 (69%)	7 (58%)	
Riding	14 (48%)	11 (31%)	5 (42%)	
Average Age	11	12	11	0.8023 ^3^
Sex				0.9508 ^2^
F	12 (41%)	13 (37%)	5 (42%)	
M	17 (59%)	22 (63%)	7 (58%)	

Key: ^1^ Fisher’s Exact Test; ^2^ Chi-squared tests; ^3^ Analysis of variance.

**Table 3 animals-13-02384-t003:** Percentages of absent left and right caudal ventral tubercles in 76 osseous specimens specifically selected with absent caudal tubercles (unilateral and bilateral) of the sixth cervical vertebra.

Grade (0–4)	0/4 (%)	1/4 (%)	2/4 (%)	3/4 (%)	4/4 (%)	Total
Left aCVT	12 (15%)	3 (4%)	6 (8%)	6 (8%)	49 (65%)	76
Right aCVT	35 (46%)	2 (4%)	4 (4.5%)	4 (4.5%)	31 (41%)	76
Total	47 (30.5%)	5 (4%)	10 (6.25%)	10 (6.25%)	80 (53%)	152

**Table 4 animals-13-02384-t004:** Percentages of left and right absent cranial ventral tubercles relative to the absence of the caudal ventral tubercle in 76 osseous specimens.

**Left aCVT Grade (*n*)**	**0/4** ***n*** = **12**	**1/4** ** *n* ** **= 3**	**2/4** ** *n* ** **= 6**	**3/4** ** *n* ** **= 6**	**4/4** ** *n* ** **= 49**
Left aCrVT Grade					
0/4	12 (100%)	3 (100%)	6 (100%)	6 (100%)	12 (24%)
1/4	0 (0%)	0 (0%)	0 (0%)	0 (0%)	9 (18%)
2/4	0 (0%)	0 (0%)	0 (0%)	0 (0%)	24 (49%)
3/4	0 (0%)	0 (0%)	0 (0%)	0 (0%)	4 (8.2%)
**Right aCVT Grade**	**0/4** ***n* = ** **35**	**1/4** ** *n* ** **= 2**	**2/4** ** *n* ** **= 4**	**3/4** ** *n* ** **= 4**	**4/4** ** *n* ** **= 31**
Right aCrVT Grade					
0/4	35 (100%)	2 (100%)	4 (100%)	4 (100%)	14 (45%)
1/4	0 (0%)	0 (0%)	0 (0%)	0 (0%)	6 (19%)
2/4	0 (0%)	0 (0%)	0 (0%)	0 (0%)	9 (29%)
3/4	0 (0%)	0 (0%)	0 (0%)	0 (0%)	2 (6.5%)

## Data Availability

All the data is declared in the manuscript.

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
