# Peer review of "Characterization of the Caudal Ventral Tubercle in the Sixth Cervical Vertebra in Modern Equus ferus caballus"

_animals, 2023, doi:10.3390/ani13142384_

Round 1
Reviewer 1 Report
The authors present an excellent review of the anatomy of the 6th cervical vertebra, correlate normal anatomy with absence of the ventral tubercle and create a clear and readily usable grading scheme. The illustrations are outstanding, well labelled and the figure legends are clear and fully explanatory. The data presentation is clear, although the bar graphs of the grades took a bit of study to understand them. The discussion regarding possible involvement of homeobox genes is highly speculative but appropriate. The assertion that absence of the ventral tubercle is a manifestation of a syndrome comparable to bovine complex vertebral malformation is not supported by any evidence in this study but the cited literature makes a strong case. The references are appropriate and all the relevant literature is cited appropriately.
I have only a few minor comments about the manuscript that I found a bit confusing.
Materials and Methods
Lines 156-156: From the ventral view of a normal C6, with the caudal aspect of the CVT aligned with the vertebral body...I assume you mean that the line drawn between the most caudal extent of the CVTs is perpendicular to the long axis of the 6th cervical body. If you mean something else, consider rewriting for clarity.
Lines 158-160: I am curious why you elected to divide the CVT and CrVT into 4 arbitrary quadrants rather than calculating the proportion of the respective tubercle that was absent, since you identified a clear landmark using the caudal edge of the transverse process. This may show the absence of the tubercle is on a continuous gradient rather than segmental.
Author Response
Dear Reviewer 1,
Thank you for your time in reviewing this manuscript. We greatly appreciate the efforts you have made in the improvement of our research and have modified the manuscript according to the recommendations.
Recommendations
The authors present an excellent review of the anatomy of the 6th cervical vertebra, correlate normal anatomy with absence of the ventral tubercle and create a clear and readily usable grading scheme. The illustrations are outstanding, well labelled and the figure legends are clear and fully explanatory. The data presentation is clear, although the bar graphs of the grades took a bit of study to understand them. The discussion regarding possible involvement of homeobox genes is highly speculative but appropriate. The assertion that absence of the ventral tubercle is a manifestation of a syndrome comparable to bovine complex vertebral malformation is not supported by any evidence in this study, but the cited literature makes a strong case. The references are appropriate, and all the relevant literature is cited appropriately.
I have only a few minor comments about the manuscript that I found a bit confusing.
Materials and Methods
COMMENT: Lines 156-156: From the ventral view of a normal C6, with the caudal aspect of the CVT aligned with the vertebral body...I assume you mean that the line drawn between the most caudal extent of the CVTs is perpendicular to the long axis of the 6th cervical body. If you mean something else, consider rewriting for clarity.
ANSWER: Reworded the sentence for clarity - From the ventral view of a normal C6, with the caudal aspect of the CVT in alignment with the caudal aspect of the vertebral body, the ventral process is divided into the CrVT and CVT as per Figure 1.
COMMENT: Lines 158-160: I am curious why you elected to divide the CVT and CrVT into 4 arbitrary quadrants rather than calculating the proportion of the respective tubercle that was absent, since you identified a clear landmark using the caudal edge of the transverse process. This may show the absence of the tubercle is on a continuous gradient rather than segmental.
ANSWER: The reasoning was to determine statistically if a certain CVT grade was indicative of an absent CrVT.

Reviewer 2 Report
Referee’s report Animals 2456920
This paper describes the variation in congenital anomalies of the 6th cervical vertebrae in a predefined selection of osseous specimens of 76 horses, known to have congenital variations.
The title does not represent the content of the paper. This paper describes variations of the ventral tubercle of C6 - both the cranial and caudal portions - and does not (& cannot) investigate functional implications.
The paper is excessively long relative to its content. The results are presented in extreme detail and some simplification and lack of repetition would make the manuscript considerably easier to read. The methods are incomplete. The Discussion is long-winded and repeats results. The Conclusions again repeat results without offering conclusions that are unique to the study or suggesting relevance.
The authors only cite references which support their thesis that such congenital variants are likely to be of clinical significance demonstrating some bias and lack of objectivity.
The lay person summary and the abstract should make it clear that only cervical vertebrae with anomalous specimens of C6 were examined.
All Table & Figure Legends should be able to be read independently from the text. It is therefore important that the legends make it clear that only horses with anomalous ventral processes of C6 were included in the study. This was not a random sample of specimens.
Line 21
‘Therefore, the findings from this study provides the clinician with a definitive grading system when reporting anomalous variations of the ventral process of C6, and especially in pre-purchase cases.’
This was not a radiographic study – you do not know if your grading system can be applied to radiographs – this is a highly misleading statement.
Line 36 What is meant by 'could create diagnostic issues associated with radiographing the caudal cervical module. '?
Keywords should provide additional words for a search engine that are not in the title.
Line 63 - what do you mean by lateral? Do you mean abaxial?
Antero-posterior is incorrect nomenclature for horses - please refer to craniocaudal and be consistent throughout the manuscript
Line 64 has (not demonstrates)
lateral-lateral view
cranial (not anterior)
caudal (not posterior)
Line 79 This is not new! See Whitwell K & Dyson S. Interpreting radiographs 8: Equine cervical vertebrae. Equine Vet. J. 1987,19: 8‑14
Line 87 Nor should it be described as fatal - there are large numbers of horses functioning normally, even at the highest level of equine sports, which have congenital anomalies of C6. It is very important that the potential significance of such anomalies is put in perspective and in the light of recent literature which demonstrate that the presence of an anomaly does not necessarily equate with dysfunction or clinically relevant disease
Line 100-2 Although clinical findings are beyond the scope of this study, the conclusions might benefit the clinician when reporting on the diagnostic images of C6 in the caudal cervical vertebrae.' This study does nothing to aid a clinician in clinical decision making - it may facilitate the correct description of radiographs - but that is not the same thing
Table 1 legend horses with
How useful is this table??
Line 133 shouldn't metric units be used?
Specify what images were acquired for each specimen
Figure 1 The position of 'Cranial' is potentially misleading; I suggest that it is moved to the top of the figure
There are some lovely images!
Line 172 Among animals (not between)
Line 173 The methods should be consistently written in the past tense
Who assessed the specimens?
Figure 3 The figure legend should make it clear which side of the specimen is being referred to
Line 183 How was the information about age, breed and work discipline determined?
Given that for inclusion in the study you had to have access to osseous collections with an anomaly of C6 what is the purpose of statistically analysing the country of origin or breed or work discipline when the numbers in each were so small? The statistics will be of low power and of questionable value.
Line 189 please clarify that this is completely absent
Line 192 This sentence is repetitive of the methods
Fig. 5 Please also describe the caudal ventral tubercles in these specimens
Line 217 - 'upon expiration' - at the time of death might be more straightforward to understand. Were the age data normally distributed? If not please use median & IQR
Table 2 This is a rather unwieldy Table; I would recommend summarising these results and publishing this table in Supplementary Information
Line 218 The inclusion of the country data and work discipline just cloud your key messages - I suggest that you omit this
Line 260 & in many places elsewhere - please avoid the repetitive use of presenting/ presented
The specimens had
Also please avoid the term animals - all you had were osseous specimens from horses
Line 261 You noted the p-value determined by the Fisher's Exact Test
Line 265 You present your data in an extremely detailed way which actually makes it quite difficult to absorb - & adds to complexity. I suggest that you make an effort to simplify it and only give the most relevant information - consider, for example, what value is it to the reader to know that of the 29 horse with bilateral aCVT, 24 were grade 4. The overall figure of 24/76 is more relevant.
Figure 7 - this adds little to what is stated in the text and moreover is repeated in Figure 9. Avoid duplication!
Line 274 'This presentation only occurred when the aCVTs graded 4/4 (complete absence of the CVT) ' - which presentation?
Line 278 Here and elsewhere please review your use of 'the' rather than a or an
Table 5 Please make it clearer in the Table legend what the p-value refers to
Line 283 'Twenty-nine animals presented bilateral morphology' I think that you mean that 24 horses had bilateral anomalous morphology of C6
Lines 285-8 This is nothing new and was described many years ago
Figure 8 - this is not new information – see Clinical Radiology of the Horse
Line 298 This is repetitive of what has already been said.
Please try to simplify your text, giving the key results without repetition, so that the information is more readily available. Extreme detail makes manuscripts difficult to read.
Line 313 the absence of what?
Lines 313 - 4 Avoid repetition of Materials and Methods or Results in the Discussion Paragraph starting line 319 Please do not muddle study design and limitations. Your study only set our to be a morphological study of osseous specimens. The results may facilitate more accurate description of radiographic images although superimposition may make interpretation challenging in some instances. Comparison between osseous specimens and radiographs would be a separate study.
The authors should make an effort to write in a less discursive and more precise way throughout the Discussion which needs to be shortened substantially.
Line 329 How would the number of horses per country benefit practitioners?
Lines 332- 343 This is completely repetitive of the Results (although more succinct) - this should not be part of a Discussion - the Discussion should focus on Discussing the results in the context of previously published work. The authors fail to make any mention of clinical studies which have questioned the clinical significance of anomalies of C6.
Line 397 Knowledge of the shape and length of C5,6 & 7 & T1 should prevent any confusion
Line 419 Repetitive of what has already been said. C7 is shorter than C6
The Conclusions are largely repetitive of both the Results and the Discussion.
What are the real conclusions - what is new from this study - that is relevant to a clinician or anyone else working in this field?
Words like 'presented' used over & over again
Confusion over use of the & a
Author Response
Dear Reviewer 2,
Thank you for your time in reviewing this manuscript. We greatly appreciate the efforts you have made in the improvement of our research and have modified the manuscript according to the recommendations.
RECOMMENDATIONS
Referee’s report Animals 2456920
This paper describes the variation in congenital anomalies of the 6th cervical vertebrae in a predefined selection of osseous specimens of 76 horses, known to have congenital variations.
COMMENT: The title does not represent the content of the paper. This paper describes variations of the ventral tubercle of C6 - both the cranial and caudal portions - and does not (& cannot) investigate functional implications.
ANSWER: Functional implications have been removed from the title.
COMMENT: The paper is excessively long relative to its content. The results are presented in extreme detail and some simplification and lack of repetition would make the manuscript considerably easier to read.
ANSWER: The length of the paper has been addressed by the removal of repetitious content. Adjustments and simplification of the results relating to the comments have been altered according to the recommendations (see below in responses).
COMMENT: The methods are incomplete. The Discussion is long-winded and repeats results. The Conclusions again repeat results without offering conclusions that are unique to the study or suggesting relevance.
ANSWER: SM-D: Have made a series of adjustments (see below).
COMMENT: The authors only cite references which support their thesis that such congenital variants are likely to be of clinical significance demonstrating some bias and lack of objectivity.
ANSWER: SM-D: I am only aware of one retrospective study that could be relevant to your above comment being the study by Veraa, S.; de Graaf, K.; Wijnberg, I.D.; Back, W.; Vernooij, H.; Nielen, M; Belt, A.J. Caudal Cervical Vertebral Morphological Variation is not Associated with Clinical Signs in Warmblood Horses. Equine Vet J 2019, 52, 210–224. Hence, I have used this study as per your comment Line 330.
COMMENT: The lay person summary and the abstract should make it clear that only cervical vertebrae with anomalous specimens of C6 were examined.
ANSWER: Adjusted
COMMENT: All Table & Figure Legends should be able to be read independently from the text. It is therefore important that the legends make it clear that only horses with anomalous ventral processes of C6 were included in the study. This was not a random sample of specimens.
ANSWER: Adjusted according to the recommendations.
COMMENT: Line 21
‘Therefore, the findings from this study provides the clinician with a definitive grading system when reporting anomalous variations of the ventral process of C6, and especially in pre-purchase cases.’
This was not a radiographic study – you do not know if your grading system can be applied to radiographs – this is a highly misleading statement.
ANSWER: This sentence has been removed and replaced with - “Further comparative studies involving radiographic images to gross observations might be of benefit to clinicians when reporting on the anomalous C6 variations.”
COMMENT: Line 36 What is meant by 'could create diagnostic issues associated with radiographing the caudal cervical module. '?
ANSWER: Removed the sentence and adjusted for clarity between lines 32 and 33. -
COMMENT: Keywords should provide additional words for a search engine that are not in the title.
ANSWER: Adjusted
COMMENT: Line 63 - what do you mean by lateral? Do you mean abaxial?
ANSWER: Made the addition of ‘left’ –
COMMENT: Antero-posterior is incorrect nomenclature for horses - please refer to craniocaudal and be consistent throughout the manuscript
ANSWER: Origins of the terminology are palaeontological and have been altered according to the nomenclature for horses.
COMMENT: Line 64 has (not demonstrates)
lateral-lateral view
cranial (not anterior)
caudal (not posterior)
ANSWER: Corrected
COMMENT: Line 79 This is not new! See Whitwell K & Dyson S. Interpreting radiographs 8: Equine cervical vertebrae. Equine Vet. J. 1987,19: 8‑14
ANSWER: Changed the context.and reference No. 32 to above.
COMMENT: Line 87 Nor should it be described as fatal - there are large numbers of horses functioning normally, even at the highest level of equine sports, which have congenital anomalies of C6. It is very important that the potential significance of such anomalies is put in perspective and in the light of recent literature which demonstrate that the presence of an anomaly does not necessarily equate with dysfunction or clinically relevant disease
ANSWER: Removed the ‘fatal’ component from the sentences.
COMMENT: Line 100-2 Although clinical findings are beyond the scope of this study, the conclusions might benefit the clinician when reporting on the diagnostic images of C6 in the caudal cervical vertebrae.' This study does nothing to aid a clinician in clinical decision making - it may facilitate the correct description of radiographs - but that is not the same thing
ANSWER: Altered the sentence to – ‘Although clinical findings are beyond the scope of this study, the conclusions might facilitate the clinician’s description of the anomalous variations in C6 when radiographed.’
MATERIALS – to clarify the selection process, the first sentence states the following – ‘To be eligible for the study, only osseous specimens of C6 from E. ferus caballus that presented an aCVT with minimal damage and clear structural definition of the remaining ventral process were selected for examination.
This statement should clarify the selection process and reporting for the rest of the study.
COMMENT: Table 1 legend horses with
How useful is this table??
ANSWER: The table allocates an identification code for each specimen, which specifically identifies it when used or discussed in the study.
It also demonstrates the congenital malformation in this study was not breed, sex or geographically dependant.
COMMENT: Line 133 shouldn't metric units be used?
ANSWER: This was an American product and written according to their specifications. Hence added the metric in brackets.
COMMENT: Specify what images were acquired for each specimen
ANSWER: Addition to Line 133/134 - Only ventral and left lateral photographs were acquired for the study.
COMMENT: Figure 1 The position of 'Cranial' is potentially misleading; I suggest that it is moved to the top of the figure
ANSWER: This adjustment has been made
COMMENT: There are some lovely images!
ANSWER: Thank you – the primary photographer was the 4th author Pamela Blades Eckelbarger.
COMMENT: Line 172 Among animals (not between)
ANSWER: Corrected to – ‘among the osseous specimens’
COMMENT: Line 173 The methods should be consistently written in the past tense
ANSWER: Corrected
COMMENT: Who assessed the specimens?
ANSWER: Addition made – ‘All C6 osseous specimens were examined by three authors; Sharon May-Davis, Diane Dzingle and Pamela Blades Eckelbarger. ‘
COMMENT: Figure 3 The figure legend should make it clear which side of the specimen is being referred to
ANSWER: Adjustments have been made to rectify the legend.
COMMENT: Line 183 How was the information about age, breed and work discipline determined?
ANSWER: When each specimen was donated for research by the owner, its history was also supplied.
COMMENT: Given that for inclusion in the study you had to have access to osseous collections with an anomaly of C6 what is the purpose of statistically analysing the country of origin or breed or work discipline when the numbers in each were so small? The statistics will be of low power and of questionable value.
ANSWER: Yes, this was discussed, and as stated in Line 336 – ‘Therefore, the number of horses and associative statistics per country/continent, breed, discipline, age, and or sex provides background information that might benefit practitioners, researchers, and or future related studies.’ Even if the information is of little relevance, the difficulty in obtaining osseous specimens with this particular congenital malformation in such numbers might provide details relevant to another researcher or study.
COMMENT: Line 189 please clarify that this is completely absent
ANSWER: Rewritten for clarity - ‘When an aCVT displays is a 4/4 grade, then the entire CVT is absent. The same incremental grading protocols apply to the CrVT.’
COMMENT: Line 192 This sentence is repetitive of the methods
ANSWER: Corrected
COMMENT: Fig. 5 Please also describe the caudal ventral tubercles in these specimens
ANSWER: Corrected
COMMENT: Line 217 - 'upon expiration' - at the time of death might be more straightforward to understand. Were the age data normally distributed? If not please use median & IQR
ANSWER: Revised – ‘The average age of the horse at the time of death was 11.3 years (median 10; interquartile 6,16).’
COMMENT: Table 2 This is a rather unwieldy Table; I would recommend summarising these results and publishing this table in Supplementary Information
ANSWER: Revised with a new summarised table 2.
COMMENT: Line 218 The inclusion of the country data and work discipline just cloud your key messages - I suggest that you omit this
ANSWER: Please refer to previous comments
COMMENT: Line 260 & in many places elsewhere - please avoid the repetitive use of presenting/ presented
The specimens had
ANSWER: Action taken.
COMMENT: Also please avoid the term animals - all you had were osseous specimens from horses
ANSWER: Adjusted throughout the manuscript.
COMMENT: Line 261 You noted the p-value determined by the Fisher's Exact Test
ANSWER: This grade presented the highest percentages in both left and right CVTs with a Fisher’s Exact Test noting a p-value=0.0013 between left and right aCVTs
COMMENT: Line 265 You present your data in an extremely detailed way which actually makes it quite difficult to absorb - & adds to complexity. I suggest that you make an effort to simplify it and only give the most relevant information - consider, for example, what value is it to the reader to know that of the 29 horse with bilateral aCVT, 24 were grade 4. The overall figure of 24/76 is more relevant.
ANSWER: Removed
COMMENT: Figure 7 - this adds little to what is stated in the text and moreover is repeated in Figure 9. Avoid duplication!
ANSWER: Removed
COMMENT: Line 274 'This presentation only occurred when the aCVTs graded 4/4 (complete absence of the CVT) ' - which presentation?
ANSWER: The presentation of aCrVTs only occurred when the aCVTs graded 4/4 (complete absence of the CVT).
COMMENT: Line 278 Here and elsewhere please review your use of 'the' rather than a or an
ANSWER: Adjusted
COMMENT: Table 5 Please make it clearer in the Table legend what the p-value refers to
ANSWER: Removed as it is written in Line 273
COMMENT: Line 283 'Twenty-nine animals presented bilateral morphology' I think that you mean that 24 horses had bilateral anomalous morphology of C6
ANSWER: Rewritten - Twenty-nine osseous C6 specimens displayed a bilateral aCVT 4/4 grade, 24/29 involved the absence of the caudal aspect of the CrVT. In this presentation, the morphology of C6 becomes representational of C5.
COMMENT: Lines 285-8 This is nothing new and was described many years ago
ANSWER: Addressed this is the discussion.
COMMENT: Figure 8 - this is not new information – see Clinical Radiology of the Horse
ANSWER: Yes agree – just describing the morphology fully for the reader as the absence of the CrVT is less recognised.
COMMENT: Line 298 This is repetitive of what has already been said.
Please try to simplify your text, giving the key results without repetition, so that the information is more readily available. Extreme detail makes manuscripts difficult to read.
ANSWER: Adjusted by the removal of previous text.
COMMENT: Line 313 the absence of what?
Lines 313 - 4 Avoid repetition of Materials and Methods or Results in the Discussion Paragraph starting line 319 Please do not muddle study design and limitations. Your study only set our to be a morphological study of osseous specimens. The results may facilitate more accurate description of radiographic images although superimposition may make interpretation challenging in some instances. Comparison between osseous specimens and radiographs would be a separate study.
ANSWER: Adjusted with addition in the Discussion of further research corresponding to the size of the aCVT and CrVT to the transposition of the CVT/CrVT onto the ventral surface of C7.
COMMENT: The authors should make an effort to write in a less discursive and more precise way throughout the Discussion which needs to be shortened substantially.
ANSWER: Substantially shortened.
COMMENT: Line 329 How would the number of horses per country benefit practitioners?
ANSWER: Removed practitioners.
COMMENT: Lines 332- 343 This is completely repetitive of the Results (although more succinct) - this should not be part of a Discussion - the Discussion should focus on Discussing the results in the context of previously published work. The authors fail to make any mention of clinical studies which have questioned the clinical significance of anomalies of C6.
ANSWER: Corrected
COMMENT: Line 397 Knowledge of the shape and length of C5,6 & 7 & T1 should prevent any confusion.
ANSWER: Christine Gee [33] made the recommendation to identify and label C5 so to overt an error. Anecdotally, a number of veterinarians are concerned about such errors and the possible associative litigation when C6 resembles C5 radiographically, as the characteristics are so variable between individuals.
COMMENT: Line 419 Repetitive of what has already been said. C7 is shorter than C6.
ANSWER: Adjusted
COMMENT: The Conclusions are largely repetitive of both the Results and the Discussion.
What are the real conclusions - what is new from this study - that is relevant to a clinician or anyone else working in this field?
Revised with reference to grades,
Comments on the Quality of English Language
Words like 'presented' used over & over again
ANSWER: Adjusted
Confusion over use of the & a
ANSWER: Found 3 in the paper and either removed or replaced with the correct terminology.
Submission Date
01 June 2023
Date of this review
12 Jun 2023 09:06:21

Round 2
Reviewer 2 Report
Referees report for R1
The revised version of this manuscript still requires further major modification before it could be considered suitable for publication.
C6 either has or does not have ventral tubercles. It does not present, display, show or exhibit or any other such word.
Punctuation and some sentence constructions need attention.
I would very strongly advise the removal of Table 1 and omitting some of the statistical analyses (see below).
Unnecessary repetition of text persists and must be removed.
The figure legends are generally considerably improved.
Lines 14 & 15 What you really mean is that only osseous specimens with congenital variants of C6 were examined
Please rephrase
Lines 21-22 This sentence reads poorly
A study comparing radiographic images with gross specimens may provide information to facilitate accurate radiological description of C6 variants.
Line 24 aCVT is not defined in the Abstract (which should be able to be read independently from the layperson summary)
Line 63 '... is caudal. These two ...'
Line 70 on either (not of)
Line 79 have would be a better word than display
I think what you actually mean is morphological change, not morphology
Line 80 remove 'in'
The case reports do not themselves identify, they describe
Lines 86-87
I think that you mean that the syndrome includes multiple axial skeleton variants
Lines 98-99 This is not relevant to the Introduction – omit
Line 110 had would be better than displayed
Line 127 & elsewhere had would be preferably to displayed
Table 1 examined would be better than observed
I see no value in this lengthy table - the data are summarised in the text – we do not need more than that
Line 135 'for the study' is unnecessary
In all figure legends C6 should be defined. Figure legends should be able to be read in isolation from the text
Line 163 has, not displays
Lines 172-3 This still does not make it clear that all specimens were examined by 3 observers. Was this done independently or concurrently.
The methods should be sufficiently precise so that the study could be repeated
The numbers of specimens from each subset (Country, breed, purpose) were too small to draw meaningful statistical conclusions. There is absolutely no point in performing statistics, other than descriptive statistics, unless you have a study with sufficient power. It is of no benefit to future researchers because the results may be completely misleading. It is not acceptable to acknowledge the low statistical power in the Discussion. The statistical analysis should simply not be done.
Line 189-191 Repeats M&M - omit. Fig. 4 could be moved to M&M
Line 193 keep things simple and correct English - had (not exhibited)
Fig. 5 has (not displays)
There is something seriously wrong about the layout of the revised Table 2 - I am unable to comment on its suitability or otherwise.
I strongly recommend that line 225 -252 are removed and the associated figures
The statistical power is inevitably low, and this level of detail detracts from the key aims of the study
Line 254 had, not exhibited
Line 264 '... 3/4; none had a 4/4 grade.
Line 264 -6 You don't need to repeatedly define your grading system. Keep the text as simple & precise as possible - then the key results are easier to extract!
Line 275 had (not displayed)
Rephrase
‘…., in 24 of which the caudal aspect of the CrVT was also absent. Therefore C6 had a similar shape to C5.’
Figure 7 All the asterixes are red, not red and black as indicated in the figure legend. The legend needs to make it clearer that c and d are C6. C5 & C6 require definitions.
Lines 282 & 3 had and had (not exhibited and displayed)
Line 284 have (not present)
Line 286 had (not displayed)
Lines 293 -8 This is very clumsily written. Please rephrase and write more succinctly. What is said is pretty meaningless, particularly because you have not compared radiological appearance with gross appearance. This is purely speculative.
Lines 309 – ‘12 crossbreds were identified with aCVT, across nine countries that stemmed both hemispheres. Hence, the postulation by DeRouen et al (2016) is likely accurate, whereby the authors described the breed association of aCVTs in differing populations suggests a genetic basis.'
I don't see how your observations support this hypothesis - you have no idea of the actual prevalence of aCVT in any of the countries in which osseous specimens were available.
Line 313 those horses which had
Lines 320 - 322 I think it is somewhat misleading to say that the cervical spine is as vertical as possible. Most resting horses have the mid neck region almost horizontal or at a slight incline. There is potentially considerable motion between individual cervical vertebrae, flexion, extension in sagittal and frontal planes and rotation
See also the work of Nicole Rombach
Line 338 I think that it must be acknowledged that it is equally likely that osteoarthritis of the articular process joints is likely to be a major contributor to pain and reduced range of motion. We know clinically that affected horses often respond favourably to medication of the joints
Lines 341 - 353 This is highly repetitive of the Introduction. Omit.
Lines 383-4 What does 'and whether certain grades are conducive to transposition onto the ventral surface of C7' actually mean?
There is no discussion about the limitations of your study and what can be concluded from it.
Line 399 data are
See above
Author Response
RE: Manuscript ID: animals-2456920 – 2nd Round
Dear Reviewer,
Thank you for your time and useful comments regarding the above manuscript.
Upon discussions with my co-authors, the statistical application has been retained with further explanation in the comments by the statistician, Elle Saber (in italics page 3/4). Hopefully, your concerns are satisfactorily met, but if not, I will leave the final decision with the Editor/s.
Overall, each comment has been answered point-by-point and trust these responses meet with your approval.
Kind Regards,
Sharon May-Davis
Referees report for R1
The revised version of this manuscript still requires further major modification before it could be considered suitable for publication.
C6 either has or does not have ventral tubercles. It does not present, display, show or exhibit or any other such word.
Punctuation and some sentence constructions need attention.
I would very strongly advise the removal of Table 1 and omitting some of the statistical analyses (see below).
Unnecessary repetition of text persists and must be removed.
Response – most of these requests have been addressed (note red font in text) or respectively argued (see below).
The figure legends are generally considerably improved.
Lines 14 & 15 What you really mean is that only osseous specimens with congenital variants of C6 were examined.
Please rephrase
Response – rephrased.
Lines 21-22 This sentence reads poorly
A study comparing radiographic images with gross specimens may provide information to facilitate accurate radiological description of C6 variants.
Response – reworded
Line 24 aCVT is not defined in the Abstract (which should be able to be read independently from the layperson summary)
Response – reworded and applied to other acronyms. This changed the Summary and Abstract to keep in line with the 200 word limit.
Line 63 '... is caudal. These two ...'
Response – reworded
Line 70 on either (not of)
Response – reworded
Line 79 have would be a better word than display
I think what you actually mean is morphological change, not morphology
Response – reworded
Line 80 remove 'in'
The case reports do not themselves identify, they describe
Response – reworded
Lines 86-87
I think that you mean that the syndrome includes multiple axial skeleton variants
Response – reworded
Lines 98-99 This is not relevant to the Introduction – omit
Response – reworded
Line 110 had would be better than displayed
Response – reworded
Line 127 & elsewhere had would be preferably to displayed
Response – reworded
Table 1 examined would be better than observed
I see no value in this lengthy table - the data are summarised in the text – we do not need more than that
Response – it has been moved to a Supplementary File.
Line 135 'for the study' is unnecessary
Response – reworded
In all figure legends C6 should be defined. Figure legends should be able to be read in isolation from the text
Response – reworded
Line 163 has, not displays
Response – reworded
Lines 172-3 This still does not make it clear that all specimens were examined by 3 observers. Was this done independently or concurrently.
Response – clarified.
The methods should be sufficiently precise so that the study could be repeated
The numbers of specimens from each subset (Country, breed, purpose) were too small to draw meaningful statistical conclusions. There is absolutely no point in performing statistics, other than descriptive statistics, unless you have a study with sufficient power. It is of no benefit to future researchers because the results may be completely misleading. It is not acceptable to acknowledge the low statistical power in the Discussion. The statistical analysis should simply not be done.
Response
I respectfully disagree, where the sample size is small Fisher’s Exact test is used. Fisher’s exact test does not rely on large sample sizes for the test statistic to be asymptotically valid, it is valid for all samples sizes (a nice summary of the two tests is provided by Kim HY. Statistical notes for clinical researchers: Chi-squared test and Fisher's exact test. Restor Dent Endod. 2017 May;42(2):152-155. doi: 10.5395/rde.2017.42.2.152. Epub 2017 Mar 30. PMID: 28503482; PMCID: PMC5426219.).
The purpose of providing a p-value with the variables in table 3 is to demonstrate that the distribution of morphological variation does not depend on the origin of the specimen, the breed (purpose) or sex. Large (non-significant) p-values quantify the level to which the data is consistent with the null hypothesis: in this case independence between the malformation type and the variable of interest. This is an important step in the analysis as it establishes the validity of the subsequent tests of dependence between grading levels. If there were a significant relationship for example if sex and the malformation side were dependent on each other, then further analysis would need to control for this.
Poor sample sizes and low power are endemic to the field of equine science, and despite this hypothesis tests are regularly presented with sample sizes much smaller than this study.
Line 189-191 Repeats M&M - omit. Fig. 4 could be moved to M&M
Response – omitted and Fig. 4 moved.
Line 193 keep things simple and correct English - had (not exhibited)
Response – reworded.
Fig. 5 has (not displays)
Response – reworded.
There is something seriously wrong about the layout of the revised Table 2 - I am unable to comment on its suitability or otherwise.
Response – issue reported to Animals MDPI. Not in revised manuscript upon submission.
I strongly recommend that line 225 -252 are removed and the associated figures
The statistical power is inevitably low, and this level of detail detracts from the key aims of the study
Response – see above.
Line 254 had, not exhibited
Response – reworded.
Line 264 '... 3/4; none had a 4/4 grade.
Line 264 -6 You don't need to repeatedly define your grading system. Keep the text as simple & precise as possible - then the key results are easier to extract!
Response – reworded.
Line 275 had (not displayed)
Response – reworded.
Rephrase
‘…., in 24 of which the caudal aspect of the CrVT was also absent. Therefore C6 had a similar shape to C5.’
Response – reworded.
Figure 7 All the asterixes are red, not red and black as indicated in the figure legend. The legend needs to make it clearer that c and d are C6. C5 & C6 require definitions.
Response – issue reported to Animals MDPI. Not in revised manuscript upon submission.
Lines 282 & 3 had and had (not exhibited and displayed)
Response – reworded.
Line 284 have (not present)
Response – reworded.
Line 286 had (not displayed)
Response – reworded.
Lines 293 -8 This is very clumsily written. Please rephrase and write more succinctly. What is said is pretty meaningless, particularly because you have not compared radiological appearance with gross appearance. This is purely speculative.
Response – reworded.
Lines 309 – ‘12 crossbreds were identified with aCVT, across nine countries that stemmed both hemispheres. Hence, the postulation by DeRouen et al (2016) is likely accurate, whereby the authors described the breed association of aCVTs in differing populations suggests a genetic basis.'
I don't see how your observations support this hypothesis - you have no idea of the actual prevalence of aCVT in any of the countries in which osseous specimens were available.
Response – reworded.
Line 313 those horses which had
Response – reworded.
Lines 320 - 322 I think it is somewhat misleading to say that the cervical spine is as vertical as possible. Most resting horses have the mid neck region almost horizontal or at a slight incline. There is potentially considerable motion between individual cervical vertebrae, flexion, extension in sagittal and frontal planes and rotation
See also the work of Nicole Rombach
Response – reworded. Incorporated ‘in a natural environment’ to clarify the point. Rombach’s 15 specimen cadaver study has been noted. Respectfully, other studies describe with specificity the range of motion relative to cervical modules, and the cranial and caudal modules provide the greatest ROM relative to the mid cervical module. From a palaeontology perspective, the ventral process of C6 is part of the mammalian body plan and has similar morphology and function in most, especially prey animals.
Line 338 I think that it must be acknowledged that it is equally likely that osteoarthritis of the articular process joints is likely to be a major contributor to pain and reduced range of motion. We know clinically that affected horses often respond favourably to medication of the joints
Response – reworded.
Lines 341 - 353 This is highly repetitive of the Introduction. Omit.
Response – reworded.
Lines 383-4 What does 'and whether certain grades are conducive to transposition onto the ventral surface of C7' actually mean?
Response – reworded.
There is no discussion about the limitations of your study and what can be concluded from it.
Response – this was omitted due to the first round review, but a small paragraph at the end of the Discussion has now been included.
Line 399 data are
Comments on the Quality of English Language - See above
